# Dependence Fidelity and Downstream Inference Stability in Generative Models

**Nazia Riasat**
North Dakota State University
`nazia.riasat@ndsu.edu`

## Abstract

Recent advances in generative artificial intelligence have led to increasingly realistic synthetic data, yet the criteria used to evaluate such models remain largely focused on marginal distribution matching and likelihood-based measures. While these diagnostics assess local realism, they provide limited insight into whether a generative model preserves the multivariate dependence structures that govern downstream inference. In this work, we argue that this gap represents a fundamental limitation in current approaches to trustworthy generative modeling. We introduce covariance-level dependence fidelity as a practical and interpretable criterion for evaluating whether a generative distribution preserves the joint structure of data beyond univariate marginals. We formalize this notion through covariance-based measures and establish three core results. First, we show that distributions can match all univariate marginals exactly while exhibiting substantially different dependence structures, demonstrating that marginal fidelity alone is insufficient. Second, we prove that dependence divergence induces quantitative instability in downstream inference, including sign reversals in population regression coefficients despite identical marginal behavior. Third, we provide positive stability guarantees, showing that explicit control of covariance-level dependence divergence ensures stable behavior for dependence-sensitive tasks such as principal component analysis. Using minimal synthetic constructions, we illustrate how failures in dependence preservation lead to incorrect conclusions in extreme-event estimation and regression despite identical marginal distributions. Together, these results highlight dependence fidelity as a useful diagnostic for evaluating generative models in dependence-sensitive downstream tasks, with implications for diffusion models and variational autoencoders, and potential extensions to large language models as future work.

## 1 Introduction

Generative models are increasingly deployed as substitutes for real data in downstream scientific and decision-making workflows (Ho et al., 2020). In many applied pipelines, a model is treated as "high-quality" if its synthetic samples are indistinguishable from the target data under marginal diagnostics: univariate goodness-of-fit tests, feature-wise summary statistics, or likelihood-based objectives that implicitly prioritize low-dimensional representations (Goodfellow et al., 2016; Kingma & Welling, 2014). This evaluation philosophy is widespread because marginal discrepancies are easy to measure, easy to visualize, and often correlate with perceptual realism (Kynkäänniemi et al., 2019; Naeem et al., 2020). However, downstream inference is rarely a marginal operation. Regression, dimensionality reduction, clustering, and representation learning depend critically on *multivariate structure* covariances, tail dependence, conditional relationships, and higher-order interactions (Jolliffe, 2002; Anderson, 2003). A generative model can therefore appear accurate under marginal audits while remaining structurally unreliable for the tasks that motivate its use.

This paper formalizes a central claim: *marginal distribution fidelity is insufficient to guarantee trustworthy downstream behavior for dependence-sensitive tasks*, whereas explicit control of covariance-level (second-order) dependence, extensions to higher-order or copula-level structure can yield quantitative stability guarantees for a broad class of dependence-sensitive tasks. We study a target data-

generating distribution $P$ on $\mathbb{R}^d$ and a candidate generative distribution $Q$, interpreted as the output of a generative AI system. Our analysis separates two questions that are often conflated in practice:

1. Representation faithfulness: do samples from $Q$ match samples from $P$ under standard distributional diagnostics?
2. Inferential trustworthiness: if $Q$ replaces $P$ in a downstream analysis, are the resulting conclusions stable and qualitatively correct?

We show that the first question cannot be answered by marginal criteria alone, and that failures of dependence preservation can induce large and even sign-reversing inferential errors that are invisible to marginal evaluations (Wasserman, 2004; D'Amour et al., 2022).

*Main results*: Our theoretical contributions establish a simple hierarchy linking distributional fidelity to downstream stability.

(i) *Marginal fidelity does not imply dependence fidelity.* We first prove an impossibility result (Theorem 1): for any dimension $d \geq 2$, there exist distributions $P$ and $Q$ whose univariate marginals match exactly, yet whose dependence structures differ. In particular, the copulas can be distinct, and covariance-level discrepancies can be made arbitrarily large through appropriate scaling constructions. This establishes that perfect marginal agreement does not constrain either linear or nonlinear dependence, and therefore cannot serve as a certificate of structural correctness.

(ii) Dependence divergence induces inferential instability. For a population linear regression task, we obtain the bound

$$|\beta(P) - \beta(Q)| \leq \frac{1}{\sqrt{2}\,\sigma_X^2} \|\Sigma_P - \Sigma_Q\|_F.$$

with equality when the covariance perturbation affects only off-diagonal entries (i.e., covariance-only perturbations under equal marginal variances).

(iii) *Dependence fidelity yields positive stability guarantees.* Finally, we show that controlling covariance-level dependence provides a sufficient condition for stability of dependence-sensitive downstream tasks. Focusing on principal component analysis (PCA), we derive bounds for both spectral stability (via Weyl-type eigenvalue perturbation) and subspace stability (via Davis–Kahan-type results (Davis & Kahan, 1963)) in terms of $\|\Sigma_P - \Sigma_Q\|_F$ and an eigengap condition (Theorem 3). Consequently, when a generative model preserves second-order dependence at an appropriate scale, it provides guarantees on the stability of key geometric properties. These guarantees are informative in the small-perturbation regime where $\|\Sigma_P - \Sigma_Q\|_F \ll \gamma$, and may become loose when the perturbation is large relative to the eigengap.

*Synthetic constructions isolating dependence effects*: To make the theory concrete, we provide minimal synthetic examples in which marginal distributions are fixed by construction while dependence is systematically altered. These examples demonstrate two practically relevant failure modes: (a) mismatched tail dependence (e.g., Gaussian copula versus $t$-copulas) leading to severe errors in joint extreme-event probabilities, and (b) correlation sign flips producing regression coefficients of opposite sign. Both constructions are invisible to marginal-only evaluation, yet they induce large downstream discrepancies.

*Implications for evaluation of generative models*: Taken together, our results motivate *dependence fidelity* as a principal criterion for trustworthy generative AI. Marginal realism does not guarantee stability for preserving multivariate structure, and thus cannot certify inferential reliability. In contrast, dependence-aware criteria including covariance-level divergence, copula-based discrepancies, and task-aligned structural constraints, provide a mathematically meaningful axis along which generative models can be audited, compared, and, in certain settings, accompanied by explicit stability guarantees.

While each mathematical component used in this paper draws on classical results from multivariate analysis and matrix perturbation theory, our contribution is conceptual and structural. We integrate these tools into a unified framework that connects distributional fidelity to inferential stability, explicitly identifying covariance-level dependence preservation as a sufficient structural condition for trustworthy stability in dependence-sensitive downstream tasks. To our knowledge, this hierarchy, linking marginal fidelity, dependence divergence, and downstream stability, has not been formalized in the generative modeling literature.

**Scope of the proposed criterion.** We emphasize that the goal of this work is not to propose a universal criterion for trustworthy generative modeling, but rather to highlight the importance of preserving dependence structure for downstream tasks that rely on second-order relationships between variables. In particular, we focus on *covariance-level dependence fidelity* as a tractable diagnostic for settings where inference procedures (e.g., regression and PCA) depend directly on covariance structure. Our results show that covariance-level control provides quantitative stability guarantees for this class of tasks, while illustrating that richer dependence criteria are needed for tasks beyond this scope.

*Roadmap*: Section 2 introduces the problem setup and formal definitions of marginal fidelity and dependence fidelity for distributions $P$ (target) and $Q$ (model). Section 3 surveys related work on generative model evaluation, distributional distances, dependence modeling, and trustworthy AI. Section 4 presents the main theoretical results establishing a hierarchy for trustworthy generative modeling. First, we show that marginal fidelity does not imply dependence fidelity (Theorem 1). Second, we quantify the downstream impact of dependence divergence by proving instability of regression-type functionals under covariance perturbations (Theorem 2). Third, we establish positive stability guarantees for dependence-sensitive tasks under explicit covariance control using spectral and subspace perturbation bounds (Theorem 3). Section 5 provides minimal synthetic constructions that isolate dependence effects, illustrating how models with identical marginals can yield qualitatively different downstream behavior. Section 6 discusses implications for the principled evaluation of generative models and for dependence-aware design in scientific and decision-making workflows.

## 2 PROBLEM SETUP AND NOTATION

Let $P$ denote a target data-generating distribution on $\mathbb{R}^d$, and let $Q$ denote the distribution induced by a generative model. We assume throughout that $P$ and $Q$ have finite second moments so that covariance matrices are well-defined. The goal of generative modeling is to approximate $P$ by $Q$ in a way that preserves not only marginal behavior, but also multivariate structure relevant to downstream statistical tasks.

**Marginal fidelity.** We say that $Q$ achieves *marginal fidelity* if all univariate marginals match:

$$P_j = Q_j, \qquad j = 1, \ldots, d,$$

where $P_j$ and $Q_j$ denote the marginal distributions of the $j$-th coordinate under $P$ and $Q$, respectively. Marginal fidelity is commonly encouraged in practice via likelihood-based objectives, moment matching, or univariate goodness-of-fit diagnostics.

**Dependence fidelity (covariance-level).** Let

$$\Sigma_P := \mathrm{Cov}_{X \sim P}(X), \qquad \Sigma_Q := \mathrm{Cov}_{X \sim Q}(X)$$

denote the covariance matrices under $P$ and $Q$. We measure covariance-level dependence discrepancy by the Frobenius distance

$$D_\Sigma(P, Q) := \|\Sigma_P - \Sigma_Q\|_F,$$

where $\|\cdot\|_F$ is the Frobenius norm. Small values of $D_\Sigma(P, Q)$ indicate that the *linear* dependence structure of $Q$ closely matches that of $P$.

We note that $D_\Sigma(P, Q)$ is scale-sensitive; in applications requiring scale invariance, correlation-based or whitened covariance comparisons may be preferable. We adopt the covariance formulation here for mathematical transparency.

**Downstream functionals and inferential stability.** We consider population-level functionals $T(\cdot)$ that depend on multivariate structure, including: (i) population regression coefficients, (ii) principal component eigenvalues and principal subspaces, and (iii) joint tail probabilities. We say that the generative model is *inferentially stable* (for a specified class of dependence-sensitive functionals) if $T(Q)$ is controlled by $T(P)$ through bounds in terms of $D_\Sigma(P, Q)$ for the class of functionals under consideration.

**Norm relations.** For any matrix $A$, the operator norm is bounded by the Frobenius norm:

$$\|A\|_2 \leq \|A\|_F.$$

Our analysis focuses on covariance-level control for mathematical transparency. This relation allows Frobenius control of covariance error to imply spectral and subspace perturbation bounds used later in the stability guarantees. In this work, dependence fidelity refers specifically to second-order (covariance-level) structure. This notion directly controls stability for linear and spectral procedures such as regression and PCA. Higher-order or nonlinear forms of dependence (e.g., copula geometry, tail dependence, or conditional structure) are not captured by this metric and represent important directions for future work.

**Clarification on dependence notions.** In this work we distinguish three related notions of dependence preservation. (i) *Covariance-level fidelity* refers to preservation of second-order structure and is the focus of our theoretical results. (ii) *Copula-level fidelity* captures nonlinear dependence and tail behavior that may not be reflected in covariance statistics. (iii) *General dependence fidelity* refers to broader structural preservation beyond second-order statistics. The formal guarantees developed in this paper apply specifically to covariance-level fidelity, while richer forms of dependence such as copula structure are discussed conceptually but fall outside the scope of the present analysis.

The next section formalizes the limitations of marginal fidelity and establishes quantitative links between dependence discrepancy and downstream instability/stability.

## 3 RELATED WORK

**Evaluation of generative models.** A large body of work evaluates generative models using likelihood-based criteria, distributional distances, or low-dimensional summary statistics (Theis et al., 2016; Sajjadi, 2018). For image and text generation, metrics such as likelihood, reconstruction error, and embedding-based distances are widely used to assess sample quality and realism (Heusel et al., 2017). While these approaches capture aspects of marginal or feature-level fidelity, they often provide limited guarantees regarding the preservation of global multivariate structure. Our results formalize this limitation in a theoretical framework by quantifying how marginal agreement can coexist with large differences in multivariate dependence and downstream behavior.

**Distributional distances and integral probability metrics.** Classical distances between distributions, including Wasserstein distance (Ramdas et al., 2015), maximum mean discrepancy (MMD) (Gretton et al., 2012), and other integral probability metrics, provide stronger notions of global distributional similarity. However, in practice these measures are frequently estimated using kernels, projections, or low-dimensional embeddings, which may implicitly emphasize marginal or local structure. Our analysis complements this literature by identifying second-order (covariance-level) dependence preservation as a minimal structural condition that directly controls the stability of common statistical functionals.

**Dependence modeling and copula theory.** The separation between marginal behavior and dependence structure is well understood in multivariate statistics through copula theory (Nelsen, 2006), which represents joint distributions via marginal transformations and a dependence component. Our work builds on this perspective but focuses specifically on second-order (covariance-level) dependence and its consequences for inference. Rather than characterizing dependence structures themselves, we quantify how deviations in covariance structure propagate to errors in downstream tasks. In particular, we derive explicit bounds linking covariance perturbations to instability in population regression coefficients and principal component analysis. This establishes a direct connection between second-order dependence preservation and inferential stability, providing an operational criterion for evaluating generative models.

**Stability of statistical procedures.** Sensitivity analysis and matrix perturbation theory have long established that many statistical procedures depend continuously on the underlying covariance structure (van der Vaart, 2000). Results such as Weyl's inequality and the Davis–Kahan theorem provide spectral and subspace stability guarantees under small perturbations. We leverage these classical

tools to derive stability bounds: when covariance-level dependence divergence is controlled, downstream functionals remain stable.

**Trustworthy and reliable AI.** Recent work on trustworthy AI has emphasized robustness, calibration, and distributional shift (D'Amour et al., 2022; Zhou, 2021). Much of this literature focuses on predictive stability under covariate changes or adversarial perturbations. Our contribution is complementary: we study the reliability of *generative* models used as data surrogates and show that trustworthiness requires preserving second-order multivariate dependence, not only marginal realism.

Taken together, these strands of work motivate the need for dependence-aware evaluation. Our results formalize *dependence fidelity* as a minimal structural requirement that connects distributional approximation to inferential reliability.

## 4 MAIN RESULTS: DEPENDENCE FIDELITY AND TRUSTWORTHY INFERENCE

We now formalize the central claim of this work: matching marginal distributions is insufficient to guarantee trustworthy behavior of generative models, while preservation of covariance-level dependence provides sufficient conditions for stability of a class of dependence-sensitive functionals, including linear regression and spectral methods.

Throughout, let $P$ denote a target data-generating distribution on $\mathbb{R}^d$ and $Q$ a generative distribution produced by a model. Our goal is to understand how discrepancies between $P$ and $Q$ at the level of dependence structure affect the reliability of downstream statistical tasks.

### 4.1 MARGINAL FIDELITY DOES NOT IMPLY DEPENDENCE FIDELITY

Current evaluation practices for generative models often focus on marginal distribution matching, either explicitly through univariate goodness-of-fit diagnostics or implicitly through likelihood-based objectives (Goodfellow et al., 2014; Theis et al., 2016; Barratt & Sharma, 2018; Borji, 2019). We first show that such criteria provide no guarantee that the joint structure of the data is preserved.

This construction exploits scale sensitivity of covariance metrics; in applications where scale invariance is desired, correlation-based or standardized covariance comparisons may provide more appropriate diagnostics.

**Theorem 1** (Marginal fidelity does not imply dependence fidelity). *Let $d \geq 2$. There exist probability distributions $P$ and $Q$ on $\mathbb{R}^d$ such that:*

1. *All univariate marginals match exactly: $P_j = Q_j$ for each coordinate $j$.*

2. *The copulas differ: $C_P \neq C_Q$,*
$$\mathcal{D}_{\mathrm{cop}}(P, Q) > 0$$
   *for any characteristic-kernel (e.g., Gaussian kernel) MMD on the copula domain.*

3. *The covariance matrices can be chosen to differ:*
$$\Sigma_P \neq \Sigma_Q, \quad \mathcal{D}_\Sigma(P, Q) := \|\Sigma_P - \Sigma_Q\|_F > 0.$$

*Moreover, the covariance divergence can be made arbitrarily large while maintaining exact marginal agreement (Nelsen, 2006; Sklar, 1959).*

This can be achieved, for example, by scaling the marginal variance. If $X \sim \mathcal{N}(0, \sigma^2)$ and the covariance matrices differ only in their correlation structure, then

$$D_\Sigma(P, Q) = \|\Sigma_P - \Sigma_Q\|_F = 2\sqrt{2}\,\sigma^2|\rho|,$$

where in the unit-variance construction of the proof ($\sigma^2 = 1$), this simplifies to $2\sqrt{2}|\rho|$ and grows without bound as $\sigma^2 \to \infty$ while the marginals remain matched.

This result establishes a fundamental limitation of marginal-based evaluation: even perfect agreement across all univariate distributions leaves the joint dependence structure unconstrained. Consequently, marginal fidelity alone cannot serve as a criterion for trustworthy generative modeling. This perspective aligns with recent discussions emphasizing task-aware evaluation of generative models, where structural properties relevant to downstream inference may be more informative than marginal distributional fit (Theis et al., 2016; D'Amour et al., 2022).

## 4.2    DEPENDENCE DIVERGENCE INDUCES INFERENTIAL INSTABILITY

Dependence mismatch is not merely a representational discrepancy; it directly affects downstream inference. We formalize this phenomenon using population linear regression as a canonical dependence-sensitive task (Anderson, 2003; van der Vaart, 2000).

Let $(X, Y) \in \mathbb{R}^2$. We assume that variables are centered (or equivalently that the regression model includes an intercept term), ensuring that the population least-squares slope is given by

$$\beta(P) := \frac{\mathrm{Cov}_P(X, Y)}{\mathrm{Var}_P(X)}.$$

The centering assumption ensures that the regression slope depends only on second-order structure, which allows us to relate inferential sensitivity directly to covariance perturbations.

**Theorem 2** (Dependence divergence directly controls inferential sensitivity.). *Let $P$ and $Q$ be distributions on $\mathbb{R}^2$ with finite second moments and matching marginal variances across $P$ and $Q$:*

$$\mathrm{Var}_P(X) = \mathrm{Var}_Q(X) = \sigma_X^2 > 0, \qquad \mathrm{Var}_P(Y) = \mathrm{Var}_Q(Y) = \sigma_Y^2 > 0.$$

*Let $\Sigma_P$ and $\Sigma_Q$ denote the covariance matrices of $(X, Y)$ under $P$ and $Q$, respectively.*

*Then*

$$|\beta(P) - \beta(Q)| \leq \frac{1}{\sqrt{2}\, \sigma_X^2} \|\Sigma_P - \Sigma_Q\|_F.$$

Equality holds under equal marginal variances AND when the covariance perturbation is confined to the off-diagonal entry.

This theorem provides a quantitative bound linking covariance-level dependence divergence to inferential error (Davis & Kahan, 1963; Anderson, 2003). In particular, two distributions with identical marginals but opposite correlation structures yield regression coefficients of opposite sign. Thus, dependence changes that are invisible to marginal diagnostics can produce large inferential discrepancies, including sign reversals. The bound is interpreted in the standard perturbation regime where the covariance perturbation is small relative to the eigengap of the reference matrix.

Under this condition, Weyl's inequality implies

$$|\lambda_k(\Sigma_P) - \lambda_k(\Sigma_Q)| \leq \|\Sigma_P - \Sigma_Q\|_2 \quad \text{for all } k.$$

Since $\|A\|_2 \leq \|A\|_F$, we obtain

$$|\lambda_k(\Sigma_P) - \lambda_k(\Sigma_Q)| \leq \|\Sigma_P - \Sigma_Q\|_F.$$

This ensures the required spectral separation between the leading eigenspace of $\Sigma_Q$ and its complement.

## 4.3    STABILITY GUARANTEES UNDER DEPENDENCE FIDELITY

While the previous results highlight the limitations of marginal-based evaluation, we next show that explicit control of dependence divergence yields positive guarantees. We consider principal component analysis (PCA) (Jolliffe, 2002; Davis & Kahan, 1963), a fundamental dependence-sensitive operation underlying representation learning and dimensionality reduction.

**Theorem 3** (Stability of PCA under covariance-level dependence fidelity). *Let $P$ and $Q$ be distributions on $\mathbb{R}^d$ with zero mean and covariance matrices $\Sigma_P$ and $\Sigma_Q$ with finite second moments.*

*(Eigenvalue stability) For each $k = 1, \ldots, d$,*

$$|\lambda_k(\Sigma_P) - \lambda_k(\Sigma_Q)| \leq \|\Sigma_P - \Sigma_Q\|_2 \leq \|\Sigma_P - \Sigma_Q\|_F.$$

*In particular,*

$$\max_{k \leq d} |\lambda_k(\Sigma_P) - \lambda_k(\Sigma_Q)| \leq D_\Sigma(P, Q).$$

*(Subspace stability) Let $U_P \in \mathbb{R}^{d \times r}$ and $U_Q \in \mathbb{R}^{d \times r}$ denote the matrices whose columns are the top-$r$ orthonormal eigenvectors of $\Sigma_P$ and $\Sigma_Q$ respectively. Let $\gamma := \lambda_r(\Sigma_P) - \lambda_{r+1}(\Sigma_P) > 0$ denote the eigengap of $\Sigma_P$. Under the small-perturbation assumption $D_\Sigma(P, Q) < \gamma$,*

$$\|\sin\Theta(U_P, U_Q)\|_2 \leq \frac{\|\Sigma_P - \Sigma_Q\|_F}{\gamma}.$$

The subspace perturbation bound is informative when the covariance discrepancy satisfies

$$D_\Sigma(P, Q) \ll \gamma,$$

where $\gamma$ denotes the eigengap of the population covariance matrix. When the eigengap is small, even minor perturbations in the covariance structure may result in substantial instability of the estimated principal subspaces. The following result shows that controlling covariance-level dependence divergence is sufficient for stability of common spectral learning procedures.

### 4.4 IMPLICATIONS FOR TRUSTWORTHY GENERATIVE AI

Taken together, Theorems 1–3 establish a clear hierarchy:

| Theorem | Claim | Key quantity |
|---------|-------|--------------|
| Theorem 1 | Marginal fidelity does not imply dependence fidelity | $D_\Sigma(P, Q)$ can be arbitrarily large with $P_j = Q_j$ |
| Theorem 2 | Dependence divergence induces inferential instability | $|\beta(P) - \beta(Q)| \leq \frac{1}{\sqrt{2}\,\sigma_X^2}\|\Sigma_P - \Sigma_Q\|_F$ |
| Theorem 3 | Dependence fidelity yields stability guarantees | $\|\sin\Theta(U_P, U_Q)\|_2 \leq D_\Sigma(P, Q)/\gamma$ |

Table 1: Summary of the three main theoretical results.

These findings motivate *dependence fidelity* as a structural diagnostic for evaluating generative models in dependence-sensitive settings (Ovadia, 2019; Zhou, 2021). Generative systems intended for downstream scientific or decision-making tasks should therefore be assessed not only for marginal realism, but also for their ability to preserve multivariate dependence structures, particularly for tasks that rely on second-order structure such as regression and principal component analysis.

## 5 SYNTHETIC EXAMPLES: ISOLATING DEPENDENCE EFFECTS UNDER PERFECT MARGINAL FIDELITY

We now present two minimal synthetic constructions designed to isolate the effect of dependence divergence while keeping all univariate marginal distributions fixed (Nelsen, 2006; Sklar, 1959). These constructions serve two complementary purposes. First, they provide concrete realizations of the structural non-identifiability established in Theorem 1. Second, they illustrate the inferential instability quantified in Theorem 2. In each construction, the marginal distributions are identical by design, ensuring that any observed discrepancy arises solely from differences in dependence structure.

### 5.1 EXAMPLE I: GAUSSIAN VS. T-COPULA — TAIL DEPENDENCE FAILURE

Our first example demonstrates that nonlinear dependence, particularly tail dependence, can vary substantially even when marginal distributions match exactly.

**Construction.** Let $(X_1, X_2)$ be a bivariate random vector with standard normal marginals. Consider two joint distributions:

- **Distribution $P$:** a Gaussian copula with correlation parameter $\rho \in (0, 1)$, corresponding to a joint normal distribution with covariance
$$\Sigma_P = \begin{pmatrix} 1 & \rho \\ \rho & 1 \end{pmatrix}.$$

- **Distribution $Q$:** a $t$-copula with the same linear correlation parameter $\rho$ but degrees of freedom $\nu > 2$ (ensuring finite variance), transformed via probability integral transforms to obtain standard normal marginals.

By construction, both $P$ and $Q$ have identical univariate marginals $N(0, 1)$ and the same copula correlation parameter $\rho$ (the dependence parameter of the copula), rather than equality of Pearson correlation of the transformed variables. Consequently, any marginal goodness-of-fit metric (e.g., KS distance) cannot distinguish between the two.

**Dependence discrepancy.** Although the marginal distributions coincide, the copulas are distinct. The Gaussian copula exhibits zero tail dependence, whereas the $t$-copula exhibits positive tail dependence. Therefore,
$$D_{\mathrm{cop}}(P, Q) := \mathrm{MMD}_k(C_P, C_Q) > 0,$$
where $\mathrm{MMD}_k$ denotes a characteristic-kernel (e.g., Gaussian kernel) maximum mean discrepancy defined on the copula space, consistent with Theorem 1 (Gretton et al., 2012). This example illustrates that dependence discrepancies beyond second-order structure can arise even when covariance and marginal properties are matched. While the theoretical results in this work focus on covariance-level dependence, the example highlights broader structural failure modes that remain invisible to marginal evaluation.

**Downstream functional.** Consider the dependence-sensitive quantity
$$T(P) := \Pr_P(X_1 > u, X_2 > u), \qquad T(Q) := \Pr_Q(X_1 > u, X_2 > u),$$
which represents the probability of a joint extreme event at threshold $u$.

**Observation.** For moderate thresholds $u$, the heavy-tail dependence of the $t$-copula yields $T(Q) \gg T(P)$, This discrepancy occurs despite perfect marginal fidelity, demonstrating that marginal realism provides no control over extreme-event behavior (Figure 1).

This example illustrates a critical failure mode for risk-sensitive applications: generative models that match marginals may still severely misrepresent joint extremes, a phenomenon invisible to marginal-only evaluation (Borji, 2019).

**Interpretation.** This example highlights a limitation of covariance-level dependence fidelity. While preservation of covariance structure ensures stability for second-order tasks such as regression and PCA, it does not capture higher-order or tail dependence. In particular, extreme-event probabilities may differ substantially even when covariance structure is preserved. This illustrates that covariance-level diagnostics address second-order stability, while richer dependence features require complementary diagnostics beyond the scope of the present analysis.

## 5.2 Example II: Correlation Sign Flip — Regression Instability

Our second example provides a direct illustration of Theorem 2, showing that linear dependence divergence alone can invert inferential conclusions.

**Construction.** Let $P$ and $Q$ be bivariate normal distributions with zero means and unit variances, differing only in the sign of correlation:
$$\Sigma_P = \begin{pmatrix} 1 & \rho \\ \rho & 1 \end{pmatrix}, \quad \Sigma_Q = \begin{pmatrix} 1 & -\rho \\ -\rho & 1 \end{pmatrix}, \quad \rho \in (0, 1).$$
Both distributions share identical marginals $N(0, 1)$.

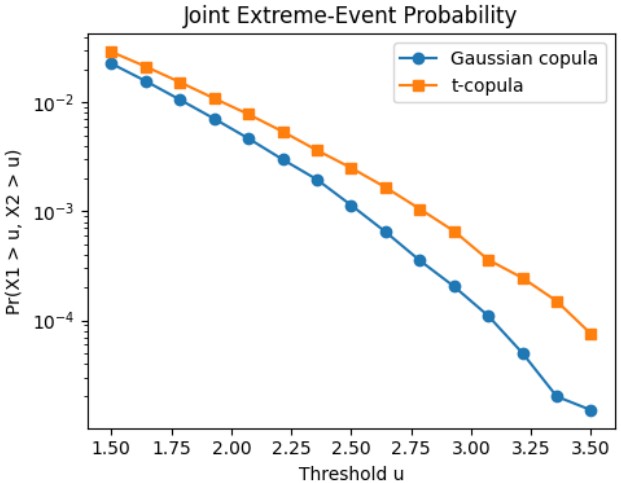

Figure 1: Joint extreme-event probabilities for Gaussian and $t$ copulas. Despite identical marginals, the $t$ copula exhibits substantially higher joint tail risk due to heavy-tail dependence.

**Downstream task.** Consider the population least-squares regression slope (Anderson, 2003)

$$\beta(P) := \frac{\text{Cov}_P(X, Y)}{\text{Var}_P(X)}.$$

Under these distributions,

$$\beta(P) = \rho, \qquad \beta(Q) = -\rho.$$

Both distributions satisfy

$$\text{Var}_P(X) = \text{Var}_Q(X) = 1,$$

so the conditions of Theorem 2 hold with $\sigma_X^2 = 1$.

**Quantitative instability.** The coefficient difference satisfies

$$|\beta(P) - \beta(Q)| = 2|\rho|,$$

while the covariance divergence is

$$\|\Sigma_P - \Sigma_Q\|_F = 2\sqrt{2}\,|\rho|.$$

Thus,

$$|\beta(P) - \beta(Q)| = \frac{1}{\sqrt{2}}\|\Sigma_P - \Sigma_Q\|_F,$$

which matches the bound in Theorem 2 with equality.

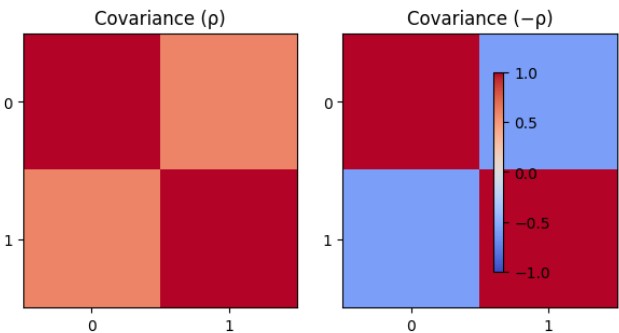

Figure 2: Covariance structures for two bivariate normal distributions with identical marginal distributions but opposite correlation signs. The change in dependence structure reverses the population regression slope, illustrating inferential instability under dependence divergence.

This example shows that even the *sign* of an inferred relationship can reverse under dependence divergence that leaves all marginals unchanged. From the perspective of scientific inference or decision-making, such sign reversals correspond to qualitatively incorrect conclusions, representing a fundamental failure of reliability. Because the covariance changes sign between $P$ and $Q$, the regression coefficient also reverses sign (Figure 2).

### 5.3 SUMMARY OF SYNTHETIC FINDINGS

Together, these examples reinforce the theoretical results of Section 4:

- Marginal fidelity does not constrain dependence (Theorem 1).
- Dependence divergence induces substantial downstream instability (Theorem 2).
- Marginal-only evaluation fails to detect these dependence-induced errors.

The constructions are intentionally simple and reproducible. Their purpose is not to provide benchmarks, but to serve as existence proofs of failure modes that any trustworthy evaluation framework for generative models must address. Complementary empirical validation on a real high-dimensional dataset is provided in Appendix B.6.

## 6  DISCUSSION: DEPENDENCE FIDELITY FOR TRUSTWORTHY GENERATIVE MODELING

The theoretical and synthetic results establish a central conclusion: for dependence-sensitive tasks, reliability is fundamentally a structural (covariance-level) property rather than a marginal one. Exact agreement of univariate distributions does not constrain multivariate dependence and therefore cannot guarantee stable downstream inference (Theorem 1). Even moderate covariance distortions can induce substantial inferential errors, including sign reversals in regression (Theorem 2), whereas explicit control of covariance divergence yields quantitative stability guarantees for procedures such as PCA (Theorem 3).

These findings highlight a limitation of current evaluation practices, which often optimize likelihood or low-dimensional summary statistics while downstream decisions depend critically on multivariate structure. Dependence mismatches may remain invisible to marginal diagnostics and cannot be mitigated simply by increasing sample size or improving marginal accuracy, reflecting a structural limitation of marginal-based evaluation (Theis et al., 2016; Borji, 2019).

Although our analysis is model-agnostic, the implications are particularly relevant for modern generative architectures. Iterative generation procedures such as diffusion models may accumulate structural distortions across steps (Ho et al., 2020; Song et al., 2021) and factorization assumptions in latent-variable models can induce dependence collapse despite accurate marginals (Locatello & Bauer, 2019). In conditional generation settings, calibrated marginal probabilities may coexist with misspecified conditional dependence.

Taken together, these results motivate *dependence fidelity* as a practical evaluation principle: models intended for downstream use should preserve task-relevant multivariate structure. In practice, covariance-level fidelity can be estimated from samples via

$$\hat{D}_\Sigma = \|\hat{\Sigma}_P - \hat{\Sigma}_Q\|_F,$$

with correlation normalization or regularization used in high-dimensional settings. This enables dependence fidelity to be incorporated as a diagnostic alongside existing evaluation metrics (Naeem et al., 2020).

**High-dimensional covariance estimation.** Reliable estimation of the covariance discrepancy $D_\Sigma(P, Q)$ from samples requires sufficient sample size relative to the ambient dimension. Under sub-Gaussian assumptions, consistent estimation of covariance matrices under the Frobenius norm typically requires $n = \Omega(d)$ samples, with empirical guidelines suggesting that $n \approx 5d$ observations often provide stable estimates in moderate dimensions (Vershynin, 2018) as a rough practical guideline. In high-dimensional regimes where $d \gg n$, naive sample covariance estimates may become

unstable. In such settings, shrinkage or regularized covariance estimators such as the Ledoit–Wolf estimator can provide reliable estimates of the underlying covariance structure. Incorporating these estimators into dependence-fidelity diagnostics represents an important direction for scaling the proposed framework to high-dimensional generative modeling applications.

**Empirical evaluation.** The present work focuses on theoretical characterization and illustrative synthetic constructions designed to isolate dependence mismatches that are invisible to marginal diagnostics. An additional empirical illustration using gene expression data is provided in Appendix B.6. Evaluating covariance-level dependence diagnostics on modern generative models (e.g., diffusion models, GAN-based generators, or tabular synthesis models) is an important next step and will be explored in future work.

## 7 CONCLUSION

This work establishes dependence fidelity as a minimal structural requirement for trustworthy generative modeling. While contemporary evaluation practices largely emphasize marginal distribution matching, our analysis shows that such criteria are insufficient to guarantee reliable downstream behavior. Even exact agreement across all marginal distributions leaves the joint structure largely unconstrained, allowing substantial distortions in multivariate dependence. We formalized the consequences of this gap through three complementary results: (i) marginal fidelity does not imply dependence fidelity (Theorem 1); (ii) covariance discrepancies induce quantifiable downstream instability, including sign reversals in regression despite identical marginal behavior (Theorem 2); and (iii) bounding covariance-level dependence divergence provides stability guarantees for dependence-sensitive procedures such as PCA (Theorem 3). These guarantees operate at the level of second-order structure and therefore apply broadly to methods whose behavior is governed by covariance geometry.

Minimal synthetic constructions demonstrated that these effects arise even in simple settings where marginal goodness-of-fit metrics are indistinguishable, yet downstream behavior differs dramatically. These findings complement recent evidence that models with well-calibrated marginals may remain structurally underspecified or unreliable for inference (D'Amour et al., 2022; Ovadia, 2019). Taken together, our results suggest that trustworthiness should be evaluated in terms of multivariate structural preservation rather than marginal agreement alone. Incorporating dependence-aware diagnostics and control provides a principal pathway toward more reliable generative systems and aligns evaluation with the requirements of scientific and decision-making applications.

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

**Appendix Overview.** This appendix provides technical details and additional experimental results supporting the theoretical claims presented in the main text. Appendix A contains complete proofs of Theorems 1–3 together with auxiliary lemmas used in the arguments. Appendix B presents synthetic experiments designed to isolate the effects of dependence structure while preserving identical marginal distributions, illustrating the phenomena described in the theoretical results. Finally, Appendix B.6 provides an empirical illustration using the Huntley Alzheimer's gene expression dataset, demonstrating that similar dependence discrepancies can arise in realistic high-dimensional biological data.

## A    APPENDIX A: PROOFS OF MAIN RESULTS

### APPENDIX A0: AUXILIARY RESULTS

This appendix provides detailed proofs of Theorems 1–3 stated in the main text. Throughout, we use the same notation as in Sections 2–4. All constructions are explicit and are chosen to isolate dependence effects while preserving identical univariate marginals.

**Lemma A.1 (Uniqueness of copula).** If a multivariate distribution has continuous marginals, its copula is unique. This follows directly from Sklar's theorem.

**Lemma A.2 (Characteristic kernels).** If $k$ is characteristic, then

$$\mathrm{MMD}_k(\mu, \nu) = 0 \iff \mu = \nu.$$

Hence distinct copulas imply strictly positive MMD.

### APPENDIX A1. PROOF OF THEOREM 1

*Proof of Theorem 1.* We construct Gaussian distributions that match exactly in all univariate marginals while differing in their multivariate dependence structure.    □

**Step 1: Construction in dimension** $d = 2$. Let $\rho \in (0, 1)$ and define two centered bivariate Gaussian distributions

$$P = \mathcal{N}(0, \Sigma_\rho), \qquad Q = \mathcal{N}(0, \Sigma_{-\rho}),$$

where

$$\Sigma_\rho = \begin{pmatrix} 1 & \rho \\ \rho & 1 \end{pmatrix}, \qquad \Sigma_{-\rho} = \begin{pmatrix} 1 & -\rho \\ -\rho & 1 \end{pmatrix}.$$

**Step 2: Exact marginal matching.** Since both covariance matrices have unit diagonal entries, each coordinate marginal under $P$ and $Q$ is $\mathcal{N}(0, 1)$. Hence, for $j = 1, 2$,

$$P_j = Q_j = \mathcal{N}(0, 1)$$

Thus the distributions agree exactly on all univariate marginals.

**Step 3: Second-order dependence mismatch.** The covariance matrices differ:

$$\Sigma_P - \Sigma_Q = \begin{pmatrix} 0 & 2\rho \\ 2\rho & 0 \end{pmatrix}.$$

Therefore,

$$D_\Sigma(P, Q) = \|\Sigma_P - \Sigma_Q\|_F = \sqrt{(2\rho)^2 + (2\rho)^2} = 2\sqrt{2}\, |\rho| > 0.$$

Hence the second-order dependence structures are different.

**Step 4: Copula mismatch.** Both $P$ and $Q$ have continuous marginals. By Sklar's theorem, each distribution admits a unique copula. Since the joint Gaussian distributions differ whenever $\rho \neq 0$, it follows that

$$P \neq Q \implies C_P \neq C_Q.$$

If $D_{\mathrm{cop}}$ is defined using a characteristic-kernel maximum mean discrepancy (MMD) on $[0,1]^2$, then

$$D_{\mathrm{cop}}(P,Q) := \mathrm{MMD}_k(C_P, C_Q) > 0,$$

Since the Gaussian copula is uniquely determined by its correlation parameter, and $\rho \neq -\rho$ whenever $\rho \neq 0$, it follows that the corresponding copulas are different.

**Step 5: Control of the dependence gap.** From Step 3, the Frobenius difference satisfies

$$D_\Sigma(P,Q) = 2\sqrt{2}\,|\rho|.$$

Thus for any $\varepsilon \in (0, 2\sqrt{2})$, choosing

$$\rho = \frac{\varepsilon}{2\sqrt{2}}$$

yields

$$\|\Sigma_P - \Sigma_Q\|_F = \varepsilon.$$

**Step 6: Extension to general $d \geq 2$.** For $d > 2$, define $P^{(d)}$ and $Q^{(d)}$ by letting the first two coordinates follow bivariate $P$ and $Q$ as constructed above, and letting the remaining coordinates be independent $\mathcal{N}(0,1)$ variables independent of the first two. Then:

- All univariate marginals of $P$ and $Q$ on $\mathbb{R}^d$ are identical.
- The covariance matrices differ only in the $(1,2)$ and $(2,1)$ entries, so

$$\|\Sigma_{P^{(d)}} - \Sigma_{Q^{(d)}}\|_F = 2\sqrt{2}\,|\rho| > 0.$$

- Since the joint distributions differ and marginals are continuous, the copulas differ, implying $D_{\mathrm{cop}}(P^{(d)}, Q^{(d)}) > 0$.

Theorem 1 establishes a fundamental limitation: exact agreement of all univariate marginals does not constrain the multivariate dependence structure. Consequently, two distributions may be indistinguishable under any marginal goodness-of-fit criterion while exhibiting substantially different joint behavior. This result formalizes why marginal fidelity alone cannot serve as a certificate of distributional realism for downstream tasks that depend on dependence structure.

APPENDIX A2. PROOF OF THEOREM 2

*Proof of Theorem 2.* Let $P$ and $Q$ be probability distributions on $(X, Y) \in \mathbb{R}^2$ with finite second moments, and assume that $\mathrm{Var}_P(X) > 0$ and $\mathrm{Var}_Q(Y) > 0$.

**Step 1: Population slope formula.** Throughout this proof we use the centering assumption of Theorem 2: $\mathbb{E}_P[X] = \mathbb{E}_P[Y] = \mathbb{E}_Q[X] = \mathbb{E}_Q[Y] = 0$. For distribution $P$, consider the population risk

$$R_P(b) := \mathbb{E}_P[(Y - bX)^2].$$

Expanding,

$$R_P(b) = \mathbb{E}_P[Y^2] - 2b\,\mathbb{E}_P[XY] + b^2\mathbb{E}_P[X^2].$$

Differentiating with respect to $b$ and setting the derivative to zero gives

$$-2\,\mathbb{E}_P[XY] + 2b\,\mathbb{E}_P[X^2] = 0,$$

hence

$$\beta(P) = \frac{\mathbb{E}_P[XY]}{\mathbb{E}_P[X^2]}.$$

By the centering assumption stated in Theorem 2, we have $\mathbb{E}_P[X] = \mathbb{E}_P[Y] = 0$. Under this assumption, the numerator satisfies

$$\mathbb{E}_P[XY] = \mathbb{E}_P[XY] - \mathbb{E}_P[X]\,\mathbb{E}_P[Y] = \mathrm{Cov}_P(X, Y),$$

and the denominator satisfies

$$\mathbb{E}_P[X^2] = \mathbb{E}_P[X^2] - (\mathbb{E}_P[X])^2 = \mathrm{Var}_P(X).$$

Therefore,

$$\beta(P) = \frac{\mathrm{Cov}_P(X, Y)}{\mathrm{Var}_P(X)}.$$

The same argument yields

$$\beta(Q) = \frac{\mathrm{Cov}_Q(X, Y)}{\mathrm{Var}_Q(X)}.$$

**Step 2: Equal-variance case.** Assume $\mathrm{Var}_P(X) = \mathrm{Var}_Q(X) = \sigma_X^2 > 0$ and $\mathrm{Var}_P(Y) = \mathrm{Var}_Q(Y) = \sigma_Y^2 > 0$. Then

$$|\beta(P) - \beta(Q)| = \left| \frac{\mathrm{Cov}_P(X, Y) - \mathrm{Cov}_Q(X, Y)}{\sigma_X^2} \right|.$$

Define

$$\Delta_{12} := \mathrm{Cov}_P(X, Y) - \mathrm{Cov}_Q(X, Y).$$

Thus,

$$|\beta(P) - \beta(Q)| = \frac{|\Delta_{12}|}{\sigma_X^2}.$$

In the unit-variance case: $\sigma_X^2 = 1$, this reduces to the bound stated in the main text.

**Step 3: Relation to covariance matrices.** Let

$$\Sigma_P = \begin{pmatrix} \sigma_X^2 & \mathrm{Cov}_P(X, Y) \\ \mathrm{Cov}_P(X, Y) & \mathrm{Var}_P(Y) \end{pmatrix}, \quad \Sigma_Q = \begin{pmatrix} \sigma_X^2 & \mathrm{Cov}_Q(X, Y) \\ \mathrm{Cov}_Q(X, Y) & \mathrm{Var}_Q(Y) \end{pmatrix}.$$

If the marginal variances are equal and the difference between $\Sigma_P$ and $\Sigma_Q$ occurs only in the off-diagonal entry, then

$$\Sigma_P - \Sigma_Q = \begin{pmatrix} 0 & \Delta_{12} \\ \Delta_{12} & 0 \end{pmatrix}.$$

The Frobenius norm satisfies

$$\|\Sigma_P - \Sigma_Q\|_F = \sqrt{\Delta_{12}^2 + \Delta_{12}^2} = \sqrt{2}\,|\Delta_{12}|.$$

Hence

$$|\Delta_{12}| = \frac{1}{\sqrt{2}}\|\Sigma_P - \Sigma_Q\|_F.$$

**Step 4: Final bound.** Substituting into the slope difference,

$$|\beta(P) - \beta(Q)| = \frac{1}{\sqrt{2}\,\sigma_X^2}\|\Sigma_P - \Sigma_Q\|_F.$$

This establishes the stated relationship between regression instability and covariance distortion. $\square$

Theorem 2 translates dependence divergence into an explicit inferential consequence. Even when marginal distributions are identical, changes in covariance structure can induce large changes in the population regression coefficient, including sign reversals. Thus, dependence mismatch directly leads to instability in downstream inference, revealing a failure mode that is invisible to marginal-based evaluation.

APPENDIX A3. PROOF OF THEOREM 3

*Proof of Theorem 3.* Let $P$ and $Q$ be distributions on $\mathbb{R}^d$ with $\mathbb{E}_P[X] = \mathbb{E}_Q[X] = 0$ and finite second moments, and define the covariance matrices

$$\Sigma_P := \mathbb{E}_P[XX^\top], \qquad \Sigma_Q := \mathbb{E}_Q[XX^\top].$$

Both $\Sigma_P$ and $\Sigma_Q$ are symmetric positive semidefinite. Recall

$$D_\Sigma(P, Q) := \|\Sigma_P - \Sigma_Q\|_F.$$

**(a) Eigenvalue stability (Weyl).** Let $\lambda_1(\Sigma) \geq \cdots \geq \lambda_d(\Sigma)$ denote the ordered eigenvalues of a symmetric matrix $\Sigma$. By Weyl's inequality for symmetric matrices, for all $k = 1, \ldots, d$,

$$|\lambda_k(A) - \lambda_k(B)| \leq \|A - B\|_2$$

.

where $\|\cdot\|_2$ denotes the operator norm.

Applying this with $A = \Sigma_P$ and $B = \Sigma_Q$ yields

$$|\lambda_k(\Sigma_P) - \lambda_k(\Sigma_Q)| \leq \|\Sigma_P - \Sigma_Q\|_2.$$

Using $\|M\|_2 \leq \|M\|_F$ for all matrices $M$, we obtain

$$|\lambda_k(\Sigma_P) - \lambda_k(\Sigma_Q)| \leq \|\Sigma_P - \Sigma_Q\|_F = D_\Sigma(P, Q).$$

Taking the maximum over $k$ yields

$$\max_{1 \leq k \leq d} |\lambda_k(\Sigma_P) - \lambda_k(\Sigma_Q)| \leq D_\Sigma(P, Q).$$

**(b) Principal subspace stability (Davis–Kahan).** Let $U_P \in \mathbb{R}^{d \times r}$ and $U_Q \in \mathbb{R}^{d \times r}$ be matrices whose columns are the top-$r$ orthonormal eigenvectors of $\Sigma_P$ and $\Sigma_Q$, respectively. By assumption (small-perturbation regime in Theorem 3), we have $D_\Sigma(P, Q) < \gamma$, which ensures the required spectral separation between the target $r$-dimensional eigenspace of $\Sigma_P$ and its orthogonal complement under perturbation by $\Sigma_Q - \Sigma_P$. This is the precise condition under which the Davis–Kahan $\sin\Theta$ theorem applies (Davis & Kahan, 1963; Yu et al., 2015). Therefore, by the Davis–Kahan $\sin\Theta$ theorem for symmetric matrices,

$$\|\sin\Theta(U_P, U_Q)\|_2 \leq \frac{\|\Sigma_P - \Sigma_Q\|_2}{\gamma}.$$

Again using $\|\Sigma_P - \Sigma_Q\|_2 \leq \|\Sigma_P - \Sigma_Q\|_F$, we conclude

$$\|\sin\Theta(U_P, U_Q)\|_2 \leq \frac{\|\Sigma_P - \Sigma_Q\|_F}{\gamma} = \frac{D_\Sigma(P, Q)}{\gamma}.$$

Combining (a) and (b), small covariance dependence divergence $D_\Sigma(P, Q)$ guarantees stability of the PCA spectrum and the leading $r$-dimensional subspace. $\square$

Theorem 3 provides a positive guarantee complementary to Theorems 1 and 2. If a generative model preserves covariance structure such that $D_\Sigma(P, Q) = \|\Sigma_P - \Sigma_Q\|_F$ is small, then the key geometric quantities used by principal component analysis remain stable. In particular, the variance explained by each principal component changes by at most $D_\Sigma(P, Q)$, and the leading PCA subspace is stable up to error $D_\Sigma(P, Q)/\gamma$, where $\gamma$ is the eigengap. Thus, covariance-level dependence fidelity is sufficient to ensure stability of dependence-sensitive downstream representations.

**Remark.** The Davis–Kahan bound is informative when $\|\Sigma_P - \Sigma_Q\|_F \ll \gamma$. If the perturbation is large relative to the eigengap, the bound may become vacuous.

# B    Synthetic Experiments Illustrating Dependence Effects

This appendix provides synthetic experiments that illustrate the mechanisms described in Theorems 1–3. All constructions use distributions with identical univariate marginals but different dependence structures, allowing us to isolate the effect of dependence divergence on joint behavior and downstream inference.

## B.1    Experimental Setup

We consider two bivariate distributions with standard normal marginals:

- $P$: a Gaussian copula with correlation $\rho$,
- $Q$: a Student-$t$ copula with the same linear correlation $\rho$ and degrees of freedom $\nu$.

The models are constructed such that

$$X_1, X_2 \sim \mathcal{N}(0, 1)$$

under both $P$ and $Q$, ensuring exact marginal agreement. Samples of size $n = 10^5$ are generated for each model. The large sample size ensures that differences reflect population-level structure rather than sampling variability. This setup isolates dependence differences arising from the copula while holding marginal distributions fixed.

## B.2    Marginal Fidelity

Figure 3 shows the empirical marginal cumulative distribution functions (CDFs) for both models. The curves are visually indistinguishable, confirming that the univariate marginals coincide by construction.

This experiment illustrates the phenomenon described in Theorem 1: agreement in all univariate marginals does not imply agreement in the joint distribution. Marginal diagnostics alone therefore fail to detect dependence differences.

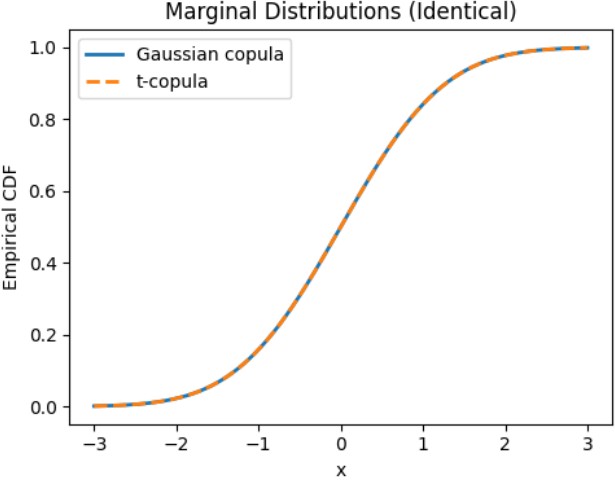

Figure 3: Empirical marginal CDFs for Gaussian and $t$-copula samples. The marginals are indistinguishable, demonstrating that marginal fidelity alone cannot detect dependence differences.

## B.3    Dependence Differences

We evaluate the dependence-sensitive functional

$$T(P) = \Pr(X_1 > u, X_2 > u),$$

introduced in Section 5.1. Although the marginal distributions coincide, the dependence structure differs markedly.

To quantify this difference, we examine joint extreme-event probabilities

$$\Pr(X_1 > u, X_2 > u)$$

across a range of thresholds $u$. The corresponding results are presented in the main text (Figure 1).

The $t$-copula exhibits substantially higher joint tail probabilities due to stronger tail dependence. This demonstrates that identical marginals can correspond to materially different joint risk profiles, consistent with Theorem 1.

## B.4 COVARIANCE-LEVEL DISTORTION

To illustrate dependence differences at the second-order level, we construct two Gaussian distributions with identical marginals but correlations of equal magnitude and opposite sign:

$$P = \mathcal{N}(0, \Sigma_\rho), \qquad Q = \mathcal{N}(0, \Sigma_{-\rho}),$$

where

$$\Sigma_\rho = \begin{pmatrix} 1 & \rho \\ \rho & 1 \end{pmatrix}.$$

The covariance matrices differ only in the off-diagonal entries, yielding the Frobenius difference

$$\|\Sigma_P - \Sigma_Q\|_F = 2\sqrt{2}\,|\rho|.$$

This construction isolates second-order dependence divergence while preserving the marginal distributions, allowing us to study the effect of covariance changes independently of marginal behavior.

## B.5 REGRESSION INSTABILITY

We examine the impact of dependence divergence on downstream inference. For each distribution, we compute the population regression slope

$$\beta = \frac{\mathrm{Cov}(X, Y)}{\mathrm{Var}(X)}.$$

Because the covariance changes sign between $P$ and $Q$, the regression coefficient also changes sign, even though the marginal distributions are identical (Figure 4). This demonstrates that identical marginals do not guarantee stability of downstream estimators. Small changes in dependence structure can induce large qualitative changes in inference, consistent with Theorem 2.

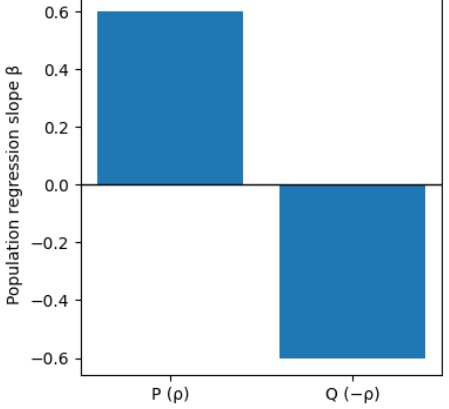

Figure 4: Population regression slope under two dependence structures with identical marginal distributions but opposite correlations. Dependence divergence induces a sign reversal in the regression coefficient, illustrating instability of downstream inference.

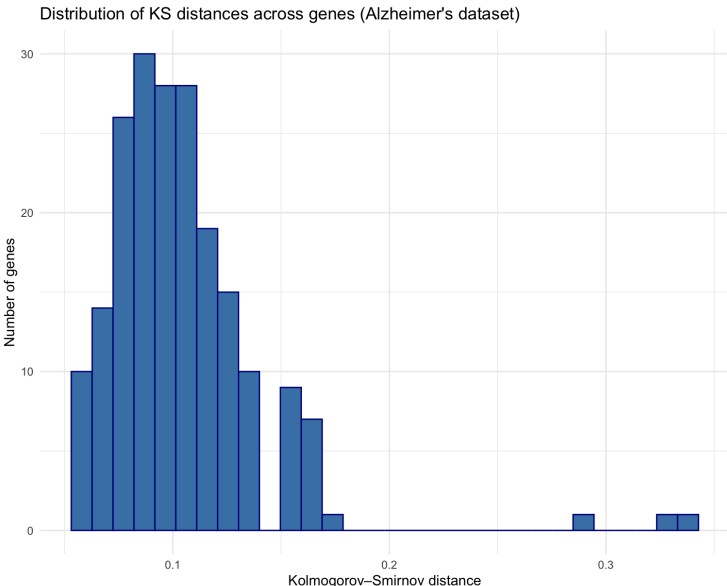

Figure 5: Distribution of Kolmogorov–Smirnov (KS) distances across genes comparing real and synthetic gene expression samples. Most KS values are small, indicating that the synthetic data approximately preserves univariate marginal distributions.

### B.6 Empirical Illustration on Gene Expression Data

To complement the synthetic constructions above, we provide a small empirical illustration using the Huntley Alzheimer's gene expression dataset available in the Gene Expression Omnibus (GEO) repository (accession: GSE125050). The dataset contains RNA-seq measurements from 113 samples of several purified brain cell types obtained from post-mortem superior frontal gyrus (SFG) tissue of Alzheimer's disease and control patients (Srinivasan et al., 2020). The goal of this experiment is not to introduce a new benchmark but to demonstrate that dependence discrepancies between real and generated data can appear even when marginal distributions appear similar. We compare real gene expression measurements with synthetic data generated from a Gaussian model fitted to the marginal distributions. We note that with $n = 113$ samples and a high-dimensional gene expression setting, this illustration falls below the $n \approx 5d$ guideline discussed in Section 6; results should therefore be interpreted as qualitative rather than as precise numerical estimates of covariance discrepancy.

**Marginal similarity (KS)**  We first evaluate marginal fidelity using Kolmogorov–Smirnov (KS) statistics across genes. Figure 5 shows the resulting histogram of KS distances indicates that most genes exhibit small marginal discrepancies, suggesting that the synthetic data approximately preserves univariate distributions.

**Covariance distortion**  Next, we examine covariance-level differences between real and synthetic data. Let $\Sigma_{\text{real}}$ and $\Sigma_{\text{syn}}$ denote the empirical covariance matrices of the real and synthetic datasets. We measure second-order dependence divergence using the Frobenius norm

$$\|\Sigma_{\text{real}} - \Sigma_{\text{syn}}\|_F.$$

Figure 6 visualizes the covariance difference matrix. Despite similar marginal behavior, noticeable covariance discrepancies remain, indicating that dependence structure is not fully preserved.

**Downstream regression sensitivity**  Finally, we examine the effect on downstream inference by comparing regression coefficients estimated from real and synthetic data (Figure 7). Although the marginal distributions are similar, the regression coefficients differ across datasets, reflecting sensitivity to underlying dependence structure.

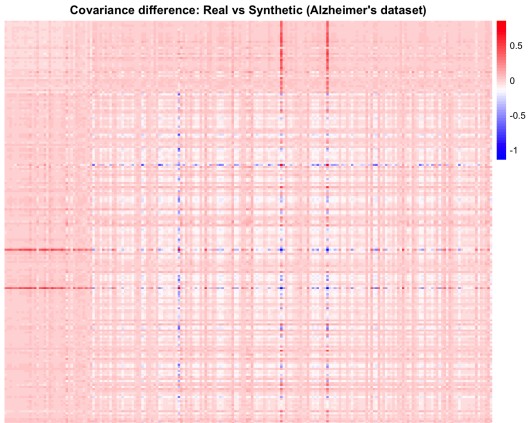

Figure 6: Heatmap of the covariance difference matrix $\Sigma_{\text{real}} - \Sigma_{\text{syn}}$ for the gene expression dataset. While marginal distributions appear similar, visible covariance discrepancies remain, illustrating differences in dependence structure between real and synthetic data.

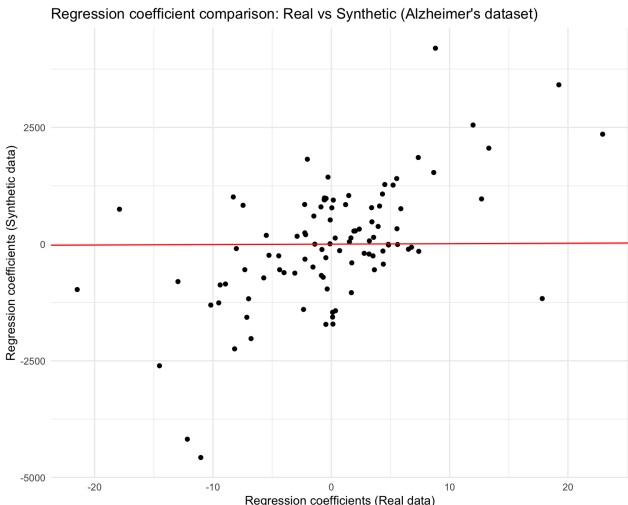

Figure 7: Comparison of regression coefficients estimated from real and synthetic gene expression data. Each point represents a coefficient estimate for a gene. Deviations from the identity relationship indicate sensitivity of downstream inference to differences in dependence structure.

## B.7 SUMMARY

These experiments collectively illustrate three key phenomena:

1. Identical marginals do not imply identical joint distributions (Theorem 1).

2. Differences in dependence structure can substantially alter joint tail risk and extreme-event behavior.

3. Covariance-level distortions can induce instability in downstream inference (Theorem 2), while small covariance divergence ensures stability for covariance-based procedures (Theorem 3).

Together, these results provide empirical illustrations for the central claim of the paper: dependence fidelity is essential for trustworthy inference, even when marginal fidelity is satisfied. The additional gene expression experiment demonstrates that the same dependence-related phenomena can arise in realistic high-dimensional biological datasets.

These experiments are constructed at the population level and therefore isolate structural effects of dependence rather than sampling variability. They illustrate that dependence distortions can remain undetected by marginal diagnostics while producing substantial changes in joint risk and downstream inference, consistent with the theoretical results.

## B.8 REPRODUCIBILITY

Code and instructions to reproduce the synthetic experiments are available at `https://github.com/NaziaRiasat/dependence-fidelity`.

