# OpenReview forum: "Dependence Fidelity and Downstream Inference Stability in Generative Models"
_mathai.club/MathAI/2026/Conference — MathAI 2026 Conference Submission_

### Official Review · Reviewer_XPaB · 2026-03-11
**A rhetorically ambitious but technically modest paper that repackages classical facts about marginals, covariance perturbations, and PCA stability without delivering genuine novelty, rigorous generative-model evaluation, or a convincing case for a new "foundational" principle.**

**Rating:** 2
**Confidence:** 5

**Review:**

# Review

## Overall assessment

**Recommendation:** Reject

This paper argues that matching univariate marginals is insufficient for trustworthy generative modeling, proposes **covariance-level dependence fidelity** as a criterion, and presents three main claims:
1. identical marginals do not imply identical dependence structure;
2. covariance mismatch can destabilize downstream inference such as regression;
3. controlling covariance perturbations yields stability guarantees for PCA.

At a very high level, the paper's central slogan is correct:
\[
(P_1,\dots,P_d) = (Q_1,\dots,Q_d) \centernot\implies \Sigma_P = \Sigma_Q.
\]
However, the manuscript is far weaker than its framing suggests. The core mathematical content is largely a repackaging of standard textbook material from copula theory, linear regression algebra, and matrix perturbation theory. The connection to **generative AI** is mostly rhetorical: there is no serious experiment on trained generative models, no empirical comparison of evaluation criteria, no finite-sample analysis, and no operational guidance beyond "also check covariance." The title and abstract promise a foundational contribution; the body delivers elementary observations and toy constructions.

## Quality

The quality is **below the bar for publication**.

The paper is built around three results that are all classical in spirit:

- The fact that matching marginals does not determine the joint distribution is standard and is one of the first lessons of multivariate statistics and copula theory.
- The regression result is essentially a one-line identity in the bivariate case:
  \[
  \beta = \frac{\operatorname{Cov}(X,Y)}{\operatorname{Var}(X)},
  \]
  so perturbing covariance perturbs the slope.
- The PCA stability result is a direct invocation of Weyl- and Davis--Kahan-type perturbation bounds.

None of this is new. The paper itself more or less admits this by saying that the mathematical components are classical and that the claimed contribution is "conceptual and structural." But even at the conceptual level, the contribution is overstated. "Marginal realism is not enough for downstream utility" is an obvious point, not a breakthrough.

Worse, the manuscript overreaches by presenting covariance-level dependence fidelity as a **foundational requirement for trustworthy generative AI**. That is not established. At best, the paper shows that second-order structure matters for a narrow family of second-order downstream procedures.

## Clarity

The paper is readable, but the clarity is **superficial rather than precise**.

The prose is repetitive and rhetorical. The same message is restated across the abstract, introduction, theorems, synthetic examples, discussion, and conclusion with minimal additional depth. Several passages sound much grander than the actual results justify.

There is also serious conceptual slippage between:

- **general dependence fidelity**,
- **copula-level dependence fidelity**, and
- **covariance-level dependence fidelity**.

These are not the same object. The paper repeatedly moves between them as if they were interchangeable. That is a major problem because the paper's own examples show they are not interchangeable.

A particularly damaging inconsistency is this: the manuscript proposes covariance-level fidelity as the practical core notion, but then highlights a **Gaussian copula vs. \(t\)-copula** example precisely to show that higher-order / tail dependence matters for extreme-event estimation:
\[
T(P) = \Pr(X_1 > u, X_2 > u).
\]
That example actually undercuts the central proposal. If the real point is that trustworthy generative modeling requires preserving tail dependence, copula geometry, and conditional structure, then covariance fidelity is plainly insufficient. If the real point is only that covariance should also be checked, the title and claims are inflated.

There are also notational and writing issues throughout. The manuscript contains multiple awkward phrases and typographical problems ("principel", confusing theorem wording, broken theorem statement syntax, vague references to characteristic kernels, etc.). These would be fixable in isolation, but here they reinforce the impression of a paper that has not been technically tightened.

## Originality

**Very low.**

The main message is a restatement of well-known facts:

- Marginals do not determine the joint law.
- Covariance perturbations affect linear regression.
- PCA is stable under small covariance perturbations.

Even the "hierarchy" the paper advertises is not meaningfully new. It is just a juxtaposition of familiar facts:
\[
\text{marginal matching} \;\not\Rightarrow\; \text{dependence matching},
\]
\[
\|\Sigma_P-\Sigma_Q\| \text{ small} \;\Rightarrow\; \text{some covariance-driven procedures are stable}.
\]

That is not a new theoretical framework; it is a standard synthesis of basic results.

The manuscript tries to claim novelty by transporting these observations into the language of "trustworthy generative AI." But relabeling classical multivariate-statistics facts with fashionable terminology is not enough. There is no new estimator, no new metric with theory, no new training objective, no nontrivial lower bound, no sample-complexity result, no benchmark, and no empirical discovery.

## Significance

**Low.**

If accepted, what would the field gain?

Not much beyond a reminder that downstream utility depends on joint structure. That reminder may be reasonable for some audiences, but it is not enough for a research contribution at conference level.

The practical recommendations are extremely thin. The paper essentially says: compute
\[
D_\Sigma(P,Q) = \|\Sigma_P - \Sigma_Q\|_F
\]
or some normalized version. But this is only a crude diagnostic, and the manuscript itself acknowledges that it is scale-sensitive, unstable in high dimensions, and blind to higher-order dependence. So even the proposed "practical and interpretable principle" is immediately weakened by the paper's own caveats.

The paper also does not demonstrate that current evaluation pipelines actually fail in the way claimed on real generative models. There is no experiment with diffusion models, VAEs, GANs, tabular synthesizers, or language models. The implications section gestures toward these models, but that is speculation, not evidence.

## Major technical concerns

### 1. The central result is trivial and does not justify the framing

Theorem 1 boils down to the fact that identical univariate marginals do not determine the copula or covariance structure. This is mathematically true but completely standard. Presenting it as a central theoretical contribution is not persuasive.

Moreover, the claim that covariance divergence can be made arbitrarily large "while maintaining exact marginal agreement" is partly a consequence of using a **scale-sensitive** metric. Inflating marginal variance inflates
\[
\|\Sigma_P - \Sigma_Q\|_F,
\]
but that says as much about the weakness of the metric as about the strength of the theorem. The paper later admits that correlation-based or whitened comparisons may be preferable. That concession substantially weakens the choice of the main proposed criterion.

### 2. Theorem 2 is mathematically under-specified and arguably mis-derived

The manuscript states the population regression slope as
\[
\beta(P)=\frac{\operatorname{Cov}_P(X,Y)}{\operatorname{Var}_P(X)}.
\]
That is the slope **with an intercept** (or after centering). However, Appendix A2 derives the expression by minimizing
\[
R_P(b)=\mathbb{E}_P[(Y-bX)^2],
\]
which is **no-intercept regression**, yielding
\[
b^\star = \frac{\mathbb{E}[XY]}{\mathbb{E}[X^2]}.
\]
These are not the same formula unless additional mean-zero assumptions are imposed. The paper's derivation silently conflates them.

This is not a cosmetic issue. It means the theorem, as written and proved, is not fully coherent. The result can likely be salvaged by explicitly assuming centered variables or by introducing an intercept properly, but the current version is sloppy.

### 3. Theorem 3 appears incomplete as stated

The PCA bound invokes Davis--Kahan in the form
\[
\|\sin \Theta(U_P,U_Q)\|_2 \le \frac{\|\Sigma_P-\Sigma_Q\|_F}{\gamma},
\]
where \(\gamma = \lambda_r(\Sigma_P)-\lambda_{r+1}(\Sigma_P)\).

As stated, this is too casual. Standard Davis--Kahan formulations require a spectral separation condition that is more delicate than merely quoting the eigengap of one matrix and dividing by it. In many versions, one needs an appropriate separation between the target eigenspace and the rest of the spectrum under perturbation, often together with a small-perturbation regime.

The paper later says the bound is "informative" when \(D_\Sigma(P,Q)\ll \gamma\), but that is not the same as stating the theorem correctly. As written, the theorem is at best incomplete and at worst misleading.

### 4. The paper's own examples contradict the advertised sufficiency of covariance fidelity

The Gaussian-copula vs. \(t\)-copula example is meant to demonstrate severe downstream errors in joint extreme-event probabilities despite identical marginals. But this example also shows that **second-order matching is not enough** for many relevant downstream tasks. In other words:

- if your goal is trustworthy inference for tail events, covariance fidelity is inadequate;
- if your goal is only trustworthy inference for covariance-driven procedures, then the paper's title and rhetoric are dramatically overstated.

The manuscript wants both messages simultaneously and does not reconcile them.

### 5. The "generative AI" connection is not earned

This is not really a paper about generative AI systems. It is a short note on basic multivariate dependence facts. There are no experiments involving trained generative models, no evidence that common generative-model metrics fail in practice for the reasons stated, and no method for improving existing models.

Saying "this matters for diffusion models, VAEs, and potentially LLMs" is not enough. The manuscript never shows:
- how these models specifically distort dependence,
- how often existing metrics miss those distortions,
- whether covariance fidelity diagnoses the failure better than alternatives,
- or how one would incorporate this into training or auditing in a principled way.

## Empirical weaknesses

The empirical section is extremely weak.

The so-called "synthetic experiments" are just toy constructions where the answer is known by design. They do not test a learned model, a real dataset, or a realistic evaluation pipeline. They merely illustrate the claims already baked into the setup.

This is especially unsatisfying because the paper repeatedly speaks in the language of **trustworthy generative AI**. If that is the target, then the paper should at minimum include something like:

- a tabular synthesis benchmark,
- a comparison of models with similar marginal diagnostics but different dependence fidelity,
- downstream-task degradation as a function of evaluation metric,
- or real-data case studies where covariance-aware checks catch failures that standard checks miss.

Without that, the paper remains a conceptual note, not a convincing AI paper.

## Missing depth

The manuscript stops exactly where the hard problems begin.

For example, it never addresses:

1. **Finite-sample estimation.**
   In practice, one only has samples from \(P\) and \(Q\). What is the concentration behavior of
   \[
   \|\widehat{\Sigma}_P-\widehat{\Sigma}_Q\|_F?
   \]
   When is this reliable in high dimension?

2. **High-dimensional structure.**
   If \(d\) is large relative to \(n\), covariance estimation is difficult. The paper mentions shrinkage and normalization in one sentence and moves on.

3. **Task alignment.**
   Why should raw Frobenius covariance error be the right metric across tasks? For regression with multiple correlated covariates, inverse covariance structure and conditioning matter. For extremes, covariance is the wrong object. For clustering or manifold learning, other geometry matters.

4. **Alternative dependence metrics.**
   If the real ambition is trustworthy evaluation, why not compare covariance-based diagnostics to copula-based distances, mutual-information preservation, multivariate two-sample tests, or downstream utility metrics?

Because the paper does not engage with any of these, it feels underdeveloped.

## Pros

- The high-level motivation is sensible: downstream utility depends on more than univariate realism.
- The paper tries to separate **representation faithfulness** from **inferential trustworthiness**, which is a useful distinction.
- The toy examples are easy to understand and may be pedagogically useful.
- The manuscript is organized in a standard theorem-example-discussion format, so the flow is easy to follow.

## Cons

- **Minimal originality**; the core results are textbook-level.
- **Overclaiming**; the title and framing promise a foundational contribution that the paper does not deliver.
- **Weak technical precision**; Theorem 2's derivation conflates regression with and without intercept, and Theorem 3 is stated too casually.
- **Conceptual inconsistency** between general dependence fidelity and covariance-level fidelity.
- **No real generative-model experiments** whatsoever.
- **No finite-sample analysis**, no high-dimensional theory, and no operational evaluation protocol.
- **Scale-sensitive main metric** that the paper itself admits may be inappropriate.
- **Repetitive writing** and several notational / typographical issues.
- **Speculative significance claims** about diffusion models, VAEs, and LLMs without evidence.

## What would be required to make this publishable

The paper would need a substantial redesign, not a light revision.

At minimum, the authors would need to:

1. **Narrow the claims drastically.**
   Stop calling this a foundational requirement for trustworthy generative AI. Call it what it is: a short note on why covariance-aware diagnostics matter for some downstream tasks.

2. **Fix the theory carefully.**
   State the regression theorem with correct assumptions, and state the PCA perturbation bound in a mathematically precise form.

3. **Resolve the core conceptual ambiguity.**
   Decide whether the paper is about:
   - covariance fidelity,
   - broader dependence fidelity,
   - or downstream-task fidelity.
   These are different notions.

4. **Provide real experiments on learned generative models.**
   Show actual models that pass common marginal or perceptual checks but fail downstream tasks due to dependence mismatch.

5. **Add finite-sample and high-dimensional analysis.**
   Otherwise the proposal is not practically actionable.

6. **Benchmark against stronger baselines.**
   Compare covariance-based diagnostics to other multivariate evaluation tools.

## Final verdict

I do not recommend acceptance.

The manuscript takes a true but elementary observation, wraps it in very broad language, and presents standard multivariate facts as if they constituted a new foundation for trustworthy generative AI. They do not. The theory is modest, the empirical evidence is essentially nonexistent, and parts of the formal development are insufficiently precise. At present, this is much closer to an expository note than to a publishable research paper.

---

### Official Review · Reviewer_vC8g · 2026-03-11
**Great theory but needs scalable real-world tests**

**Rating:** 5
**Confidence:** 4

**Review:**

The paper identifies a critical blind spot in generative AI evaluation: the over-reliance on marginal distribution matching. By formalizing "covariance-level dependence fidelity" , the authors provide a rigorous theoretical hierarchy linking distributional approximation to downstream inferential stability.The theoretical foundation is excellent. The authors apply classical multivariate statistics to prove that perfect marginal fidelity does not ensure joint dependence fidelity (theorem 1). Furthermore, they cleanly demonstrate that covariance divergence explicitly degrades linear regression (theorem 2) , while bounding it stabilizes spectral methods like PCA (theorem 3).However, the paper fails to deliver on the "Generative AI" promise of its title. The empirical validation is strikingly weak, relying entirely on minimal bivariate ($d=2$) synthetic distributions, such as Gaussian versus t-copulas. There is absolutely no evaluation of actual deep generative models on real-world datasets. Additionally, relying on the Frobenius norm of covariance differences, defined as $D_{\Sigma}(P,Q) := ||\Sigma_P - \Sigma_Q||_F$ , poses severe scalability and estimation challenges in high-dimensional settings, an issue the authors only address in passing.The authors must significantly upgrade the empirical section.

 I recommend the following improvements:
1) Evaluate the proposed metric on state-of-the-art generative models ( diffusion or GANs) against standard real-world benchmarks to prove practical utility.
2) Rigorously address the sample complexity of estimating high-dimensional covariance discrepancies, perhaps by exploring shrinkage estimators as briefly mentioned in the text.

In its current state, this is a mathematically statistics paper but an incomplete machine learning submission.

---

### Decision · Program_Chairs · 2026-03-17

**Decision:**

Accept (Poster)

**Comment:**

Dear Author(s),

On behalf of the Program Committee of the International Conference on Mathematics of Artificial Intelligence (MathAI 2026), we are pleased to inform you that your paper has been accepted for a poster presentation at MathAI 2026.

Your paper was evaluated through a rigorous two-stage review process involving both automated screening and expert review by members of the Program Committee. The reviewers recognized the quality and contribution of your work.

Important Note: The reviewers have recommended final revisions to your manuscript before the conference. Please ensure that all reviewer comments are carefully addressed in your camera-ready version. We trust that you will complete these revisions before the conference deadlines.

Presentation details:

    Format: Poster presentation

    Mode: You may present either in person (offline) at the conference venue in Sirius, Russia, or remotely via Zoom. Please indicate your preferred mode when confirming your participation.

    Conference dates: March 30 - April 3, 2026

    Website: https://mathai.club

Next steps:

    Please confirm your participation and presentation mode by replying to this email (mathai.club@yandex.ru) no later than March 15, 2026 18:00 Moscow time.

    If you plan to attend in person, the organizing committee will provide accommodation details separately.

    Please prepare your final camera-ready manuscript according to the formatting guidelines available at https://mathai.club and upload it to OpenReview by March 15, 2026 18:00 Moscow time. Ensure that all reviewer feedback has been incorporated into this final version.

Should you have any questions regarding the program, logistics, or your presentation, please do not hesitate to contact us.

We look forward to your contribution to MathAI 2026.

With kind regards,

MathAI 2026 Program Committee
International Conference on Mathematics of Artificial Intelligence
https://mathai.club
OpenReview: https://openreview.net/group?id=mathai.club/MathAI/2026/Conference
Telegram: https://t.me/MathAI_club
Email: mathai.club@yandex.ru